# Optimizing type, date, and dose of compost fertilization of organic cotton under climate change in Mali: A modeling study

Ginette Dembélé[1], Romain Loison🔾[2,3,4]*, Amadou Traoré[1], Sidiki Gabriel Dembélé[5], Fagaye Sissoko[6]

1 Regional Agricultural Research Center, Institute of Rural Economy (IER), Sotuba, Bamako, Mali, 2 CIRAD, UPR AIDA, Cotonou, Bénin, 3 AIDA, CIRAD, Univ Montpellier, Montpellier, France, 4 IRC, Cotonou, Bénin, 5 Rural Polytechnic Institute for Training and Applied Research (IPR/IFRA), Katibougou, Koulikoro, Mali, 6 National Committee for Agricultural Research (CNRA), Bamako, Mali

* romain.loison@cirad.fr

**Data Availability Statement:** All relevant data are within the manuscript and its Supporting Information files.

## Abstract

Adapting organic farming to climate change is a major issue. Cotton yields in Mali are declining due to deteriorating climatic conditions, soil fertility, and poor management. This study aimed to improve organic cotton yield in Mali in the future climate with the optimal choice of compost type, date, and dose of application. Experimental data collected in 2021 from the Sotuba research station in Mali was used for calibration and evaluation of the crop model DSSAT CSM-CROPGRO-Cotton model using phenology, leaf area index, and seed cotton yield. Climate data from the RCP4.5 and RCP8.5 scenarios of the GFDL-ESM2M model were used for future weather datasets for 2020-2039, 2040-2059, and 2060-2079. The model was able to simulate anthesis and maturity with excellent results, with nRMSE < 4%, and seed cotton yields moderately well, an nRMSE of 26% during calibration and 20.3% in evaluation. The scenario RCP8.5 from 2060 to 2079 gave the best seed cotton yields. Seed cotton yields with RCP4.5 and RCP8.5 were all better with the mid-May application period of small ruminant silo compost at 7.5 t/ha. In such conditions, more than 75% of the cases would produce more than 2000 kg/ha of seed cotton.

## 1. Introduction

Climate change remains one of our world's greatest challenges with all its implications (insufficient, excessive, or poorly distributed rainfall and rising temperatures) for the environment and the global economy [1]. From the 1970s onwards, we have witnessed a reduction in agricultural production worldwide due to the impact of climate change on rainfall and temperature [2, 3]. Climate forecasts for the African continent show endemic climatic hazards [4]. The RCP4.5 and RCP8.5 scenarios show that temperatures will gradually rise in southern Mali [5]. Negative consequences on crop productivity are the corollary of rising temperatures [5].

The origins of organic farming can be traced back to the 1920s and 1930s in Northern Europe [6]. Preserving and improving soil health is at the heart of organic farming [7].

**Funding:** This research was carried out with the financial support of the project "Appui à la Transition Agroécologique en zone Cotonnier du Mali" (AgrECo): AgrECo CML1430 whose financing agreement has been signed between AFD and the Government of Mali. The funders had no role in study design, data collection and analysis, decision to publish, or preparation of the manuscript.

**Competing interests:** The authors have declared that no competing interests exist.

Numerous studies have shown that organic farming is more effective at preserving or improving soil quality [8–10]. In developing countries, especially Mali, conventional cotton growers are in crisis due to declining soil fertility, input costs, resistant pests, or low cotton prices [11]. Many farmers are turning to organic farming to restore soil fertility, promote ecological regulation, and reduce production costs [12]. Compost, in particular, has been found helpful in restoring soil quality in Sub-Saharan Africa [13]. Organic manure can reduce soil organic carbon loss [14] and should be encouraged to reduce the impact of climate change [15].

In West and Central Africa, the agricultural industry is dominated by cotton growing in many countries. It remains the primary source of income, with sixteen million people involved in processing and marketing [16]. Agricultural development of cotton is strongly polarized between three paradigms: the agro-industrial, genetically modified, and agroecological models [17]. Conventional production requires synthetic chemical inputs that harm the environment and soil [18–21]. In this context, agroecological and fair trade initiatives using exclusively organic manure have emerged as exciting alternatives [22, 23]. Seed cotton yields vary according to climatic conditions and agronomic practices such as nutrient fertility management [24, 25]. In West Africa, the recommendation for applying organic manure to fields remains unique and independent of the quality of organic manure available on farms [26]. A low amount of compost did not induce short-term increases in seed cotton yields in Benin [27]. Also in Benin, seedcotton yields responded to different types and doses of compost, with obtained yield ranging between those observed under non-fertilized and mineral fertilizer conditions [28]. In Mali's cotton-growing zone, organic fertilizer is generally applied between March and April [29]. This period seems to expose nutrients to various losses, especially nitrogen, which remains the reference element in organic fertilizers. Understanding this practice will help determine the best time to apply organic inputs to maximize seed cotton yields.

For optimal crop yields, understanding the impacts of climate change is essential for good planning and appropriate responses [30–32]. Modeling studies could help to forecast and minimize different production constraints with a global picture considering different aspects of crop production [33]. Modeling is used to understand better these complex links between the pedoclimatic context and the responses of the agroecosystem, but also for decision-support purposes. Small-scale field experiments can be used by modeling to predict larger-scale phenomena, including the impact of likely future climate, so practical experimentation and modeling become complementary [34, 35]. Modeling using DSSAT platform models is adapted to better understand complex systems between pedoclimatic and agroecosystems for scenario analyses that will be used to support decision-making [36, 37]. Data used for model calibration and validation include initial soil conditions, crop-specific cultivation techniques, and seasonal weather conditions; these data are used by cropping system models (CSM) [38]. The CSM has been used for crop management in different environments, studies of the impact of climate change on production [39, 40], and to simulate the effects of climate change on agricultural production in Africa [41].

The impact of compost on yield and soil properties has been extensively studied. However, more research is needed to examine the specific practices used by organic cotton growers and the environmental aspects of those practices [22]. Natural resources are scarce, and mobilizing them in sufficient amounts is difficult for large-scale organic farming [42]. Adequate soil fertility management practices are essential to ensure efficient and sustainable use in the future [43]. However, Mali has no manure management policies or responsible ministries dealing with such questions yet [44]. More literature is needed on the impact of dose, type, and timing of application on cotton yield and development in Sub-Saharan rainfed cotton under current and future climates. Therefore, the objective of this study is to identify optimal use of compost on organic cotton production in Mali under expected climate change conditions to improve

Malian organic cotton farmers' resilience to climate change and provide ministries with evidence for putative compost management policies.

## 2. Materials and methods

### 2.1 Study area

This research was carried out in 2021 at the IER (Institute of Rural Economy) research station in Sotuba (12.6581°N, -7.92316°W, 320 m). Sotuba's climate is Sudano-Sahelian. It is subdivided into a rainy season from June to October and a dry season from November to May. The rainy season peaks around July and August. In 2021, precipitation reached 991 mm on 102 rainy days (Fig 1).

The soil was Sandy loam (0-20cm) and Loam (20-40cm) with low overall fertility (Table 1).

### 2.2 Field experiment

**2.2.1 Experimental design.** The experimental design was a split plot with three factors and four replicates. Each elementary plot was 12.8 m$^2$, with four lines of 4 m each, and observations were made on 5.44 m$^2$ on the two central lines by removing 2 x 0.3 m of border. The soil was evenly ploughed with a Massey Ferguson disc sprayer. The previous crop was

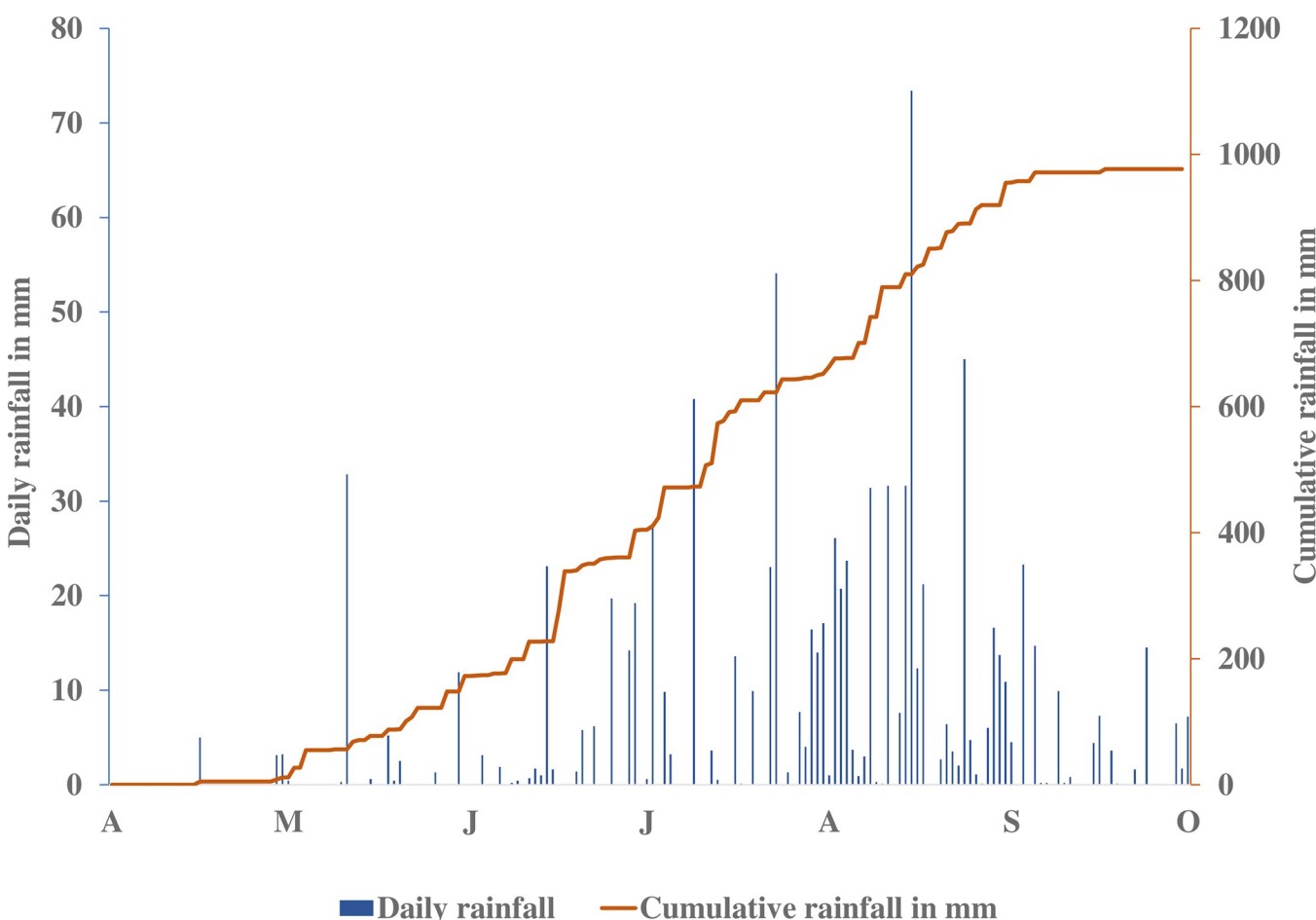

**Fig 1. Rainfall at the Sotuba research station in 2021.** A = April, M = May, J= June, J = July, A = August, S= September, O: October.

**Table 1. Soil properties of the experimental field before compost application.**

| Soil property | 0-20 cm | 20-40 cm |
|---|---|---|
| pH | 5.57 | 6.04 |
| *SOC (%) | 0.62 | 0.21 |
| N total (%) | 0.06 | 0.05 |
| Phosphorus assimilable (ppm) | 25.41 | 9.17 |
| *CEC (meq/100g) | 7.65 | 7.31 |
| Ca exchangeable (meq/100g) | 3.24 | 3.13 |
| Mg exchangeable (meq/100g) | 1.62 | 1.57 |
| K exchangeable (meq/100g) | 0.44 | 0.83 |
| Na exchangeable (meq/100g) | 0.58 | 0.5 |
| Sand (%) | 64 | 50 |
| Silt (%) | 28 | 31 |
| Clay (%) | 8 | 19 |
| Textural class | Sandy loam | Loam |
| Bulk density (g/cm3) | 1.74 | 1.55 |

*Cation exchange capacity (CEC), Soil organic carbon (SOC)

*Brachiaria ruziziensis.* The cotton (*Gossypium hirsutum* L) cultivar used was NTA MS334. This cultivar is widely cultivated in the various cotton-growing areas of Mali. It has a 120-day cycle and an average conventional seed cotton yield of about 1,500 kg/ha. The cotton was planted on July 01, 2021, with 80 cm between rows and 30 cm between two plants in the row. Pests were controlled with extracts of *Cassia nigricans.* The cotton was harvested manually on October 30, 2021.

**2.2.2 Factors studied.** Three factors related to compost were studied. In the split-plot design, the main plot factor was the period of compost application with three levels. The application was on March 01, 2021 (application date P1), on April 01, 2021 (application date P2), and on May 15, 2021 (application date P3). The second factor was the type of compost. Four types of compost were used: Silo compost based on small ruminant manure (sheep and goats) mixed with bush straw (compost C1), Silo compost based on manure from large ruminants (cattle) mixed with bush straw (compost C2), farmer compost based on small ruminant manure (compost C3) and farmer's compost based on large ruminant manure (C4). The composts C1 and C2 consistently had higher N, P, and K content than C3 and C4 (Table 2).

The third factor was the compost dose applied, especially to the line. Three doses were applied: 2.5 t/ha (dose D1), the recommended dose of 5 t/ha (dose D2), and 7.5 t/ha (dose D3). The second and third factors were fully randomized within the main plots.

**Table 2. Compost properties on the production site before March application in 2021 and 2022.**

| Compost | Content in 2021 (%) | | | Content in 2022 (%) | | |
|---|---|---|---|---|---|---|
| | N | P₂O₅ | K₂O | N | P₂O₅ | K₂O |
| C1 | 1.01 | 11.69 | 0.36 | 1.22 | 9.60 | 0.52 |
| C2 | 1.13 | 10.05 | 0.41 | 1.31 | 11.15 | 0.53 |
| C3 | 0.82 | 2.41 | 0.26 | 0.92 | 4.68 | 0.30 |
| C4 | 0.70 | 0.74 | 0.25 | 0.68 | 3.45 | 0.29 |

The composts were Silo compost based on small ruminant manure (sheep and goats) + bush straw (compost C1), Silo compost based on manure from large ruminants (cattle) + bush straw (compost C2), farmer compost based on small ruminant manure (compost C3), farmer's compost based on large ruminant manure (C4).

**2.2.3 Data collected.**    Data was collected on weather, soil, crop management and crop phenology, leaf area index (LAI), and yield. Weather data were collected from the Sotuba weather station (Auget Basculant rain gauge, Campbell Stokes Casella London heliograph, Lambrecht Meteo GmbH anemometer, Jules Richard Instruments) located near the experimental plots. It measured daily global radiation, maximum and minimum temperatures, wind speed, relative humidity, and rainfall. The soil types encountered were hydro-morphic alluvial soils and ferruginous soils with a sandy loam texture. They were low in organic matter (Nitrogen = 0.06% in the 0-20 cm horizon and 0.05% in the 20-40 cm horizon, organic carbon = 0.62% in the 0-20 cm horizon and 0.42% in the 20-40 cm horizon) with a pH of 5.57 in the 0-20 cm horizon and 6.04 in the 20-40 cm horizon and Bulk density was 1.74 mg m$^{-3}$ in the 0-20 cm horizon and 1.54 mg m$^{-3}$ in the 20-40 cm horizon.

The crop phenology, LAI, and seed cotton yield were measured. Regarding phenology, the flowering date (anthesis) is the first day on which the cumulative number of white flowers equals or exceeds half the number of plants. The date of boll opening (maturity) is the first day on which the number of open bolls on both lines is equal to or greater than half the number of plants. The LAI was measured with the LICOR LAI-2200C every two weeks from day 30 to 130 after planting to day 130 to determine the maximum LAI. The seed cotton yield was obtained from a manual harvest of each plot on October 30, 2021.

## 2.3 Crop model CSM-CROPGRO-Cotton

**2.3.1 General description.**    The DSSAT platform was developed by an international network of scientists working on the "International Benchmark Sites Network for Agrotechnology Transfer" project [45]. The DSSAT platform is a decision-support tool that integrates soil, climate, and crop management knowledge. It is used by researchers, educators, consultants, producers, and policy-makers for a variety of purposes, including crop management, climate change impact studies, precision agriculture, and sustainable research [45, 46]. The model CSM-CROPGRO-Cotton from DSSAT version 4.8 was used for the study [39, 47, 48]. This model simulates the functioning of the crop in the plot; it considers the soil with four sub-modules (soil water, soil temperature, soil carbon and nitrogen, and soil dynamics) [46], climatic conditions (temperature, average relative humidity, average wind, solar radiation, and daily rainfall), cotton crop "genetic" characteristics and crop management [49]. Phenology, especially time from planting to flowering (ADAP) and to maturity (MDAP), the maximum leaf area index (LAIX), and seed cotton yield (HWAM) are critical outputs of the DSSAT models used by most agri-modelers [50]. The model from DSSAT is sensitive to compost application, and residue retention significantly affected the simulation of cumulative N mineralization, SOC %in Africa [51]. Moreover, the CSM-CROPGRO-Cotton has been extensively used for cotton evaluation under current climate and climate change conditions in West and Central Africa [52–54].

**2.3.2 Calibration and evaluation of CSM-CROPGRO-Cotton.**    Calibration involves adjusting model parameters to best reproduce site-specific conditions [49]. Evaluation involves assessing the model's predictive capacity with an independent dataset. The model CSM-CROPGRO-Cotton has been extensively calibrated and evaluated for cotton under sub-Saharan African conditions [40, 52, 53]. Nevertheless, a specific calibration and evaluation were performed under an organic cotton production system in Mali for the study.

The CSM-CROPGRO-Cotton model was calibrated using 12 cropping conditions under compost application date P1 and evaluated using 24 independent cropping conditions at P2 and P3 compost application dates (Table 3). We calibrated the model using flowering date, boll opening, maximum LAI, and seed cotton yield to obtain reasonable estimates of the model's genetic coefficients for the cultivar used.

**Table 3. Data used for the model calibration and evaluation.**

| Use | Time of application | Dose | Type | Treatment |
|---|---|---|---|---|
| Calibration | March (P1) | D1 | C1 | P1 D1 C1 |
| | | | C2 | P1 D1 C2 |
| | | | C3 | P1 D1 C3 |
| | | | C4 | P1 D1 C4 |
| | | D2 | C1 | P1 D2 C1 |
| | | | C2 | P1 D2 C2 |
| | | | C3 | P1 D2 C3 |
| | | | C4 | P1 D2 C4 |
| | | D3 | C1 | P1 D3C1 |
| | | | C2 | P1 D3 C2 |
| | | | C3 | P1 D3 C3 |
| | | | C4 | P1 D3 C4 |
| Evaluation | April (P2) | D1 | C1 | P2 D1 C1 |
| | | | C2 | P2 D1 C2 |
| | | | C3 | P2 D1 C3 |
| | | | C4 | P2 D1 C4 |
| | | D2 | C1 | P2 D2 C1 |
| | | | C2 | P2 D2 C2 |
| | | | C3 | P2 D2 C3 |
| | | | C4 | P2 D2 C4 |
| | | D3 | C1 | P2 D3 C1 |
| | | | C2 | P2 D3 C2 |
| | | | C3 | P2 D3 C3 |
| | | | C4 | P2 D3 C4 |
| | Mid-May (P3) | D1 | C1 | P3 D1 C1 |
| | | | C2 | P3 D1 C2 |
| | | | C3 | P3 D1 C3 |
| | | | C4 | P3 D1 C4 |
| | | D2 | C1 | P3 D2 C1 |
| | | | C2 | P3 D2 C2 |
| | | | C3 | P3 D2 C3 |
| | | | C4 | P3 D2 C4 |
| | | D3 | C1 | P3 D3 C1 |
| | | | C2 | P3 D3 C2 |
| | | | C3 | P3 D3 C3 |
| | | | C4 | P3 D3 C4 |

The composts were Silo compost based on small ruminant manure (sheep and goats) + bush straw (compost C1), Silo compost based on manure from large ruminants (cattle) + bush straw (compost C2), farmer compost based on small ruminant manure (compost C3), farmer's compost based on large ruminant manure (C4); The doses were 2.5 t/ha (dose D1), the recommended dose of 5 t/ha (dose D2) and 7.5 t/ha (dose D3).

**2.3.3 The goodness of fit statistics.** The model was tested against measured data on flowering date, the beginning of boll opening (maturity) date, LAI, and cotton seed cotton yield. The statistics used to evaluate the performance of the DSSAT model were the root mean square error (RMSE) [55] and the normalized RMSE (nRMSE, in %). Model simulations have been considered excellent, good, fair, and poor based on the respective normalized RMSE (nRMSE,

Eq 1) of <10%, 10-20%, 20-30%, and >30% [56, 57].

$$nRMSE(\%) = \frac{RMSE}{observed\ mean} \times 100 \qquad (Eq1)$$

## 2.4 Virtual experiment

**2.4.1 Climate scenarios used.** Climate change projections from the Coupled Model Inter-comparison Project Phase 5 (CMIP5) GFDL-ESM2M model [58] were used to generate three weather series of 20 years each: 2020-2039, 2040-2059 and 2060-2079. Two Representative Concentration Pathways (RCP) were considered. RCP4.5 is an optimistic scenario in which emissions reach stability before the end of the 21st century at a low level [59], and RCP8.5 is a pessimistic scenario in which emissions continue to rise throughout the 21st century [60]. The dynamic of atmospheric concentration of $CO_2$ was accounted for in each scenario (Fig 2). At the end of the century, the atmospheric concentration of $CO_2$ will increase under RCP8.5 with a mean temperature of 3 to 6°C and stabilize under RCP4.5 with a mean temperature of 2 to 5°C [4].

**2.4.2 Model simulations with future climate.** Three future periods were used: 2020-2039, 2040-2059 and 2060-2079. Contrasted compost dates of application were used for the simulation experiment (P1 and P3), doses (D1 and D3), and type of compost (C1 vs C3) to simulate the future climate data under the two emission scenarios (RCP4.5 and RCP8.5). The variables evaluated were the date to flowering (days after planting, ADAP), the LAIX, and HWAM at the optimum planting date, i.e., the planting date that gave the best seed cotton yield simulated over a range of planting dates of 160, 170, 180, 190, 200 and 210 days of the year (from June 09 to July 27).

## 3. Results

### 3.1 Calibration and evaluation results

The starting coefficients for cotton cultivar NTA MS334 (ecotype CO0006) were 48.0, 40, 170, 300, and 1.10 for EM-FL, SD-PM, SLAVR, SIZLF, and LFMAX, respectively (please refer to Table 4 for the description of the coefficients). These original coefficients were those for the cultivar Deltapine 1219. They had led to high levels of RMSE on phenology, LAI, and seed cotton yield (not shown). After the calibration, the calibrated parameters values were 47.7, 20, 249.5, 245, and 0.66, respectively, for EM-FL, SD-PM, SLAVR, SIZLF, and LFMAX.

The nRMSE of ADAP was excellent (1.1%) on a basis with a mean of 62 days after planting (DAP) and an RMSE of 0.7 with the calibration data (Table 5), and it was 3.2% with a mean of 62 and 63.8 DAP respectively and an RMSE of 2.1 with the evaluation data. Similarly, the calibration was excellent for MDAP with an nRMSE of 3.8%, and evaluation was excellent with an nRMSE of 0.4%. Calibration was good for LAIX, with an nRMSE of 14.7%. However, the model underestimated LAI most of the time, and hence the evaluation could have been better, with an nRMSE of 34.7%. The calibration and evaluation of seed cotton yield were fair, with nRMSE of 26.0% and 20.3%, respectively. The calibrated average values for HWAM were 1047 kg/ha (measured) and 992 kg/ha (simulated). At the same time, the evaluated values were 890 kg/ha (measured) and 820 kg/ha (simulated).

### 3.2 Future weather data

Under the scenario RCP4.5, the climate forecast for 2040-2059 was generally rainy with an average rainfall of 1,042.8 mm, while 2020-2039 was characterized by low rainfall of 956.5 mm. Temperatures were the highest during 2060-2079 (20.72°C TMIN and 41.38°C TMAX), and

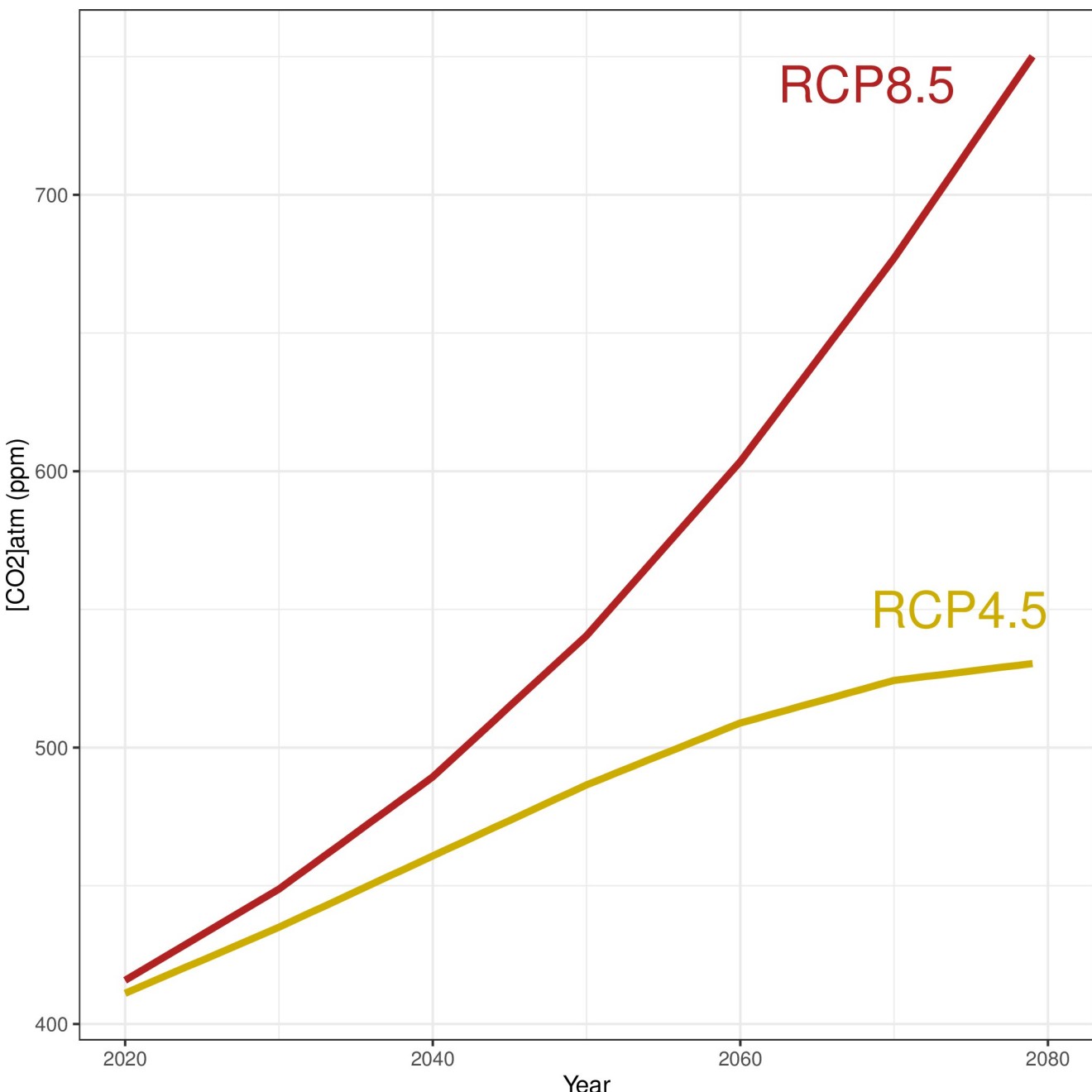

**Fig 2. Atmospheric CO$_2$ from 2020 to 2080 under scenarios RCP4.5 and 8.5 (adapted from [1]).**

solar radiation was higher (20.59 MJ m$^{-2}$ on average) compared to 2020-2039. The solar radiation for 2040-2059 was lower, with an average of 20.27 MJ m$^{-2}$ and a TMAX of 40.74˚C and TMIN of 20.57˚C. The highest minimum and maximum temperatures were in May, respectively, with 23.5˚C and 45.1˚C (2020-2039), 24.2˚C and 45˚C (2040-2059), and 24.1˚C and 46.0˚C (2060-2079) (Table 6).

Under the scenario RCP8.5, the average climate forecast for the 2020-2039 period had heavy rainfall of 1015.4 mm, followed by 2040-2059 (993.2 mm) and 2060-2079 (929 mm). Temperature was highest in the 2060-2079 (22.6˚C TMIN and 40.5˚C TMAX). Whereas, on

**Table 4. Calibrated genetic "cultivar" coefficients of the cotton cultivars as default and for cultivar NTA MS334.**

| Code | Parameter full description | Default value | Calibrated value |
|---|---|---|---|
| EM-FL | The time between cotton emergence and appearance of these flowers [photothermal day] | 48.0 | 47.7 |
| SD-PM | Time from cotton first seed (R5) and physiological maturity (R7) [photothermal day] | 40 | 20 |
| SLAVR | Specific leaf area of cultivar under standard growth conditions [cm$^2$/g] | 170.0 | 249.5 |
| SIZLF | Maximum size of a leaf [cm$^2$] | 300 | 245 |
| LFMAX | Maximum leaf photosynthesis rate at 30˚C, 350 vpm CO2 and high light levels [mg CO2/m$^2$/s] | 1.10 | 0.66 |
| Ecotype | Code for the ecotype to which this cultivar belongs | CO0006 | CO0006 |

average, solar radiation was higher in the 2060-2079 period (21.0 MJ m$^{-2}$) compared to the 2020-2039 (20.5 MJ m$^{-2}$) and 2040-2059 (20.4 MJ m$^{-2}$) periods (Table 7).

### 3.3 Future climate under organic cotton cultivation

Irrespective of the RCP scenario or the period, the treatments did not influence the distribution of ADAP (Fig 3). Generally, the ADAP was between 57 and 66 DAP. Flowering should happen earlier under RCP8.5 compared to RCP4.5.

Under the RCP4.5 scenario over 2020-2039, 50% of ADAP (interquartile Q1-Q3) were between 62 and 63 DAP. During 2040-2079, flowering occurred earlier; more than 75% of ADAP occurred before 63 DAP. The period 2060-2079 had more variability than the first two periods, characterized by ADAP up to 66 DAP, with 50% of plants flowering between 63 and 66 DAP.

Under RCP8.5, the distribution of ADAP varied between periods—the furthest the period, the faster the flowering. Between 2020 and 2039, 75% of flowering took place between 60 and 63 DAP. In contrast, between 2040 and 2059, ADAP had less variability, all falling within three days between 59 and 62 DAP. As for the 2060-2079 period, at least 75% of the beginning of flowering happened up to 60 DAP.

The distribution of LAIX differed between treatments, RCP scenarios, and periods (Fig 4). There is a clear difference between treatments over all the simulation periods. The distribution showed more variability of LAIX under RCP4.5 compared to under RCP8.5 for 2040 to 2079. The mid-May treatment period using 7.5 t/ha of small ruminant silo compost (P3C1D3) had the highest median LAIX across situations ranging from approximately 4.55 to 6.45. Conversely, the P1C3D3 produced the smallest LAIX, ranging from approximately 2.40 to 3.85.

The distribution of seed cotton yields differed between treatments, RCP scenarios, and periods (Fig 5). The range of median seedcotton yield is 610 to 2403 kg/ha. The variability of

**Table 5. CSM-CROPGRO-Cotton calibration and evaluation results for NTA MS334 at Station de Recherche Agronomique de Sotuba, Mali.**

| Variable | Unit | Calibration | | | | Evaluation | | | |
|---|---|---|---|---|---|---|---|---|---|
| | | Obs | Sim | RMSE | nRMSE (%) | Obs | Sim | RMSE | nRMSE (%) |
| ADAP | DAP | 62.0 | 62.0 | 0.7 | 1.1 | 63.8 | 62.0 | 2.1 | 3.2 |
| MDAP | DAP | 103.0 | 107.0 | 4.0 | 3.8 | 107.0 | 107.0 | 0.5 | 0.4 |
| LAIX | - | 2.8 | 2.8 | 0.4 | 14.7 | 2.7 | 2.4 | 0.9 | 34.7 |
| HWAM | kg/ha | 1 047 | 992 | 272 | 26.0 | 890 | 820 | 181 | 20.3 |

ADAP is the duration between planting and the beginning of flowering, MDAP is the duration between planting and the start of boll opening, LAIX is the maximum leaf area index, and HWAM is the seed cotton yield. Obs is the observed mean, and Sim is the simulated mean. DAP: Days after planting. RMSE is the relative mean square error, and nRMSE is the normalized RMSE (see M&M section for extensive description).

**Table 6. Average monthly climate of projected meteorological data (2020-2079) from RCP4.5 of GFDL-ESM2M model (CMIP5) at the Agronomic Research Station of Sotuba, Mali.**

| Month | SRAD (2020-2039) | SRAD (2040-2059) | SRAD (2060-2079) | TMIN (2020-2039) | TMIN (2040-2059) | TMIN (2060-2079) | TMAX (2020-2039) | TMAX (2040-2059) | TMAX (2060-2079) | Rainfall (2020-2039) | Rainfall (2040-2059) | Rainfall (2060-2079) |
|---|---|---|---|---|---|---|---|---|---|---|---|---|
| Jan. | 18.4 | 18.8 | 18.8 | 15.1 | 15.4 | 15.5 | 36.9 | 37.7 | 37.9 | 0.9 | 0.9 | 1.0 |
| Febr. | 21.2 | 21.4 | 21.3 | 17.7 | 18.7 | 18.6 | 40.4 | 40.9 | 42.0 | 0.2 | 0.0 | 0.0 |
| Mar | 22.8 | 22.4 | 22.6 | 21.0 | 21.2 | 22 | 42.8 | 43.4 | 43.2 | 2.1 | 3.0 | 2.6 |
| April | 22.3 | 23.1 | 22.6 | 23.1 | 23.8 | 24.1 | 44.2 | 45.5 | 45.0 | 31.0 | 26.5 | 37.2 |
| May | 21.7 | 21.8 | 21.4 | 23.5 | 24.2 | 24.1 | 45.1 | 45.8 | 46.0 | 65.6 | 65.3 | 69.7 |
| June | 21.2 | 19.7 | 19.8 | 22.4 | 22.8 | 23.2 | 42.3 | 42.6 | 42.4 | 119.1 | 153.5 | 171.2 |
| July | 21.1 | 20.1 | 20.6 | 21.5 | 21.7 | 21.9 | 40.5 | 39.5 | 40.0 | 206.1 | 252.1 | 245.4 |
| Aug. | 19.6 | 18.8 | 19.9 | 20.9 | 21.4 | 21.3 | 38.4 | 37.4 | 39.3 | 294.2 | 275.3 | 261.4 |
| Sept. | 19.5 | 18.4 | 20.9 | 21.1 | 21.5 | 21.5 | 38.3 | 38.1 | 41.0 | 174.5 | 201.2 | 163.9 |
| Oct. | 20.5 | 20.1 | 20.8 | 20.2 | 20.8 | 21.0 | 40.6 | 40.5 | 41.8 | 54.9 | 59.9 | 68.3 |
| Nov. | 20.3 | 20.6 | 20.2 | 17.3 | 17.6 | 18.0 | 39.9 | 40.0 | 41.2 | 7.9 | 5.1 | 12.3 |
| Dec | 18.6 | 18.0 | 18.2 | 17.0 | 17.7 | 17.4 | 36.3 | 37.5 | 36.8 | 0.0 | 0.0 | 0.0 |
| **Total** | **20.6** | **20.3** | **20.6** | **20.1** | **20.6** | **20.7** | **40.5** | **40.7** | **41.4** | **956.5** | **1 042.8** | **1 033** |

SRAD: global solar radiation, TMIN: minimum temperature, TMAX: maximum temperature, Jan: January, Feb: February, Mar: March, Aug: August, Sept: September, Oct: October, Nov: November, Dec: December, Total: the average of monthly averages.

seedcotton yields is reduced with time, especially under RCP8.5. The furthest the period, the lower the median of seed cotton yield under RCP4.5, whereas under RCP4.5, it was the contrary. Under both RCPs, the most profitable treatment is the mid-May application of small ruminant silo compost at 7.5 t/ha (P3C1D3), with median seed cotton yields ranging from 1608 to 2403 kg/ha.

**Table 7. Average monthly climate of projected meteorological data (2020-2079) from RCP8.5 of GFDL-ESM2M model (CMIP5) at the Agronomic Research Station of Sotuba, Mali.**

| Month | SRAD (2020-2039) | SRAD (2040-2059) | SRAD (2060-2079) | TMIN (2020-2039) | TMIN (2040-2059) | TMIN (2060-2079) | TMAX (2020-2039) | TMAX (2040-2059) | TMAX (2060-2079) | Rainfall (2020-2039) | Rainfall (2040-2059) | Rainfall (2060-2079) |
|---|---|---|---|---|---|---|---|---|---|---|---|---|
| Jan. | 18.8 | 18.3 | 18.9 | 15.5 | 15.9 | 18.4 | 37.5 | 38.4 | 38.7 | 1.3 | 0.9 | 1.6 |
| Febr. | 21.3 | 21.2 | 21.2 | 18.1 | 18.9 | 21 | 40.6 | 41.2 | 42 | 0.4 | 0.1 | 0.3 |
| Mar | 22.8 | 22.1 | 22.9 | 21.1 | 22.7 | 23.7 | 41.8 | 42.6 | 43.2 | 1.1 | 3.9 | 2 |
| April | 23.1 | 22.6 | 22.8 | 22.8 | 24.6 | 26 | 43.2 | 43.7 | 44.6 | 18.4 | 23.9 | 18.3 |
| May | 21.2 | 20.6 | 21.1 | 23.7 | 24.7 | 26.1 | 41.9 | 41.9 | 43.4 | 73.9 | 79.5 | 71.6 |
| June | 21.2 | 20.6 | 20.9 | 22.8 | 23.4 | 25 | 40 | 40.1 | 41.4 | 124.7 | 130.8 | 126.1 |
| July | 20.9 | 20.9 | 24.3 | 21.4 | 22.1 | 23.8 | 37.2 | 37.3 | 39.3 | 223.4 | 205.2 | 184.1 |
| Aug. | 19.6 | 20.1 | 22.8 | 21.3 | 21.8 | 22.5 | 36.8 | 36.6 | 39.2 | 288.2 | 280.3 | 248.6 |
| Sept. | 19.1 | 20 | 20.5 | 21.4 | 22 | 22.5 | 35.5 | 36.3 | 37.9 | 200.3 | 182.9 | 174 |
| Oct. | 19.2 | 19.6 | 18.7 | 20.5 | 21 | 22.8 | 36.6 | 37.5 | 38.9 | 74.9 | 77.1 | 88.8 |
| Nov | ember 20 | 19.8 | 19.5 | 17.5 | 18.5 | 19.7 | 35.9 | 37.2 | 39.4 | 8.8 | 8.6 | 13.6 |
| Dec | 18.3 | 18.4 | 18.7 | 17.1 | 18.2 | 19.2 | 36.6 | 37.2 | 38.2 | 1.3 | 0 | 1.3 |
| **Total** | **20.5** | **20.4** | **21.0** | **20.2** | **21.15** | **22.6** | **38.6** | **39.2** | **40.5** | **1015.4** | **993.2** | **929** |

SRAD: global solar radiation, TMIN: minimum temperature, TMAX: maximum temperature, Jan: January, Feb: February, Mar: March, Aug: August, Sept: September, Oct: October, Nov: November, Dec: December, Total: the average of monthly averages.

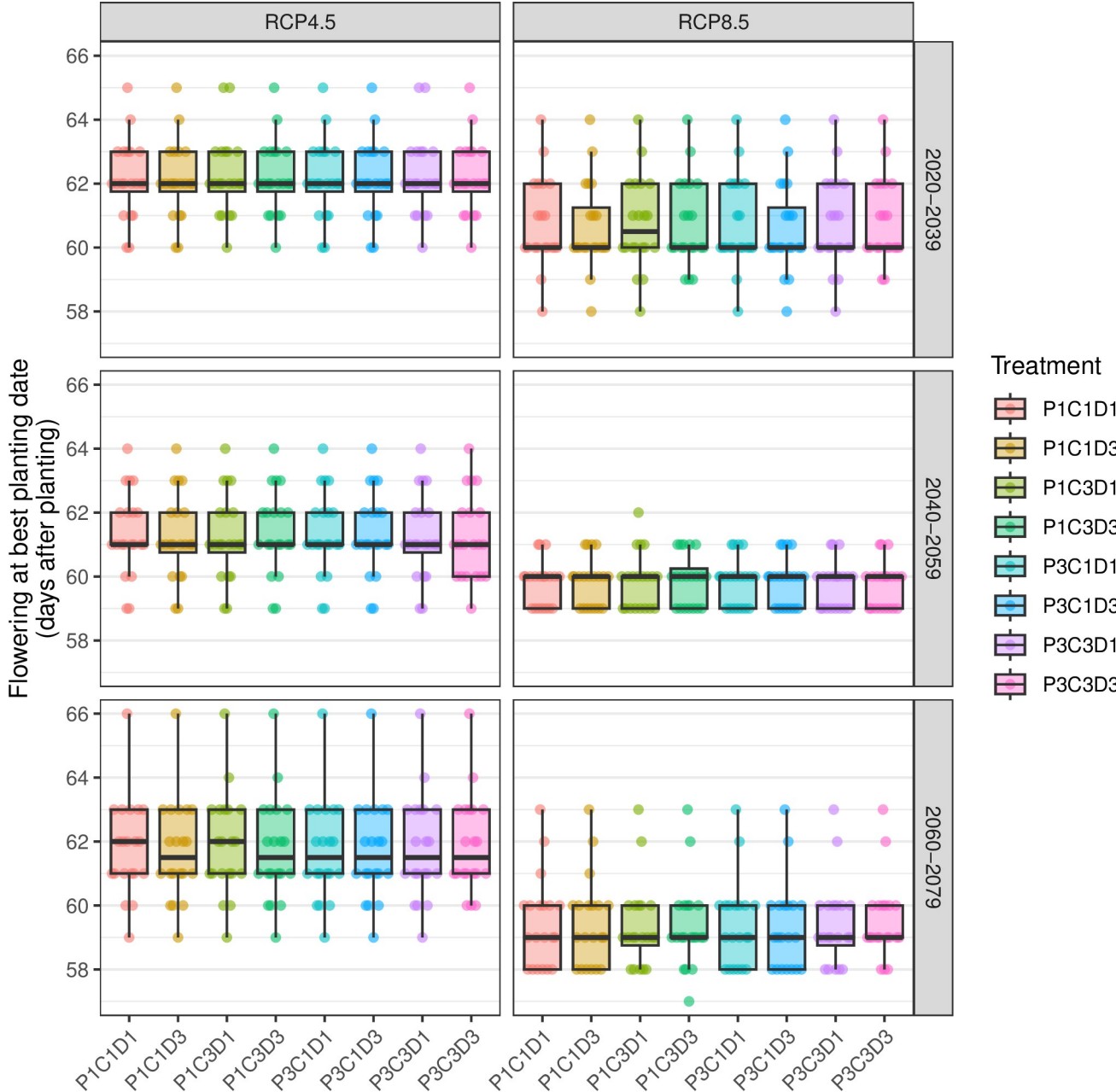

**Fig 3. Impact of future climate on flowering according to RCP, period, and practices linked to compost fertilization.** Compost applied in March (P1) and in mid-May (P3), Silo compost based on small ruminant manure (sheep and goats) + bush straw (C1), Farmer compost based on small ruminant manure (C3), dose of compost applied of 2.5 t/ha (D1) and 7.5 t/ha (D3). For each boxplot, the points are the individual values (n = 20).

## 4. Discussion

Our study used the CSM-CROPGRO-Cotton model from the platform DSSAT to calibrate and evaluate organic farming with different compost management practices in Mali. Calibration and evaluation of the model were good, with RMSE of 0.7 and 2.1 days for flowering, RMSE of 3.95 and 0.46 days for simulated boll opening, and yield with RMSE of 272 kg/ha (calibration) and 181 kg/ha (evaluation). Our results are consistent with those of [52], where calibration and evaluation of growth and yield were sufficient with low levels of RMSE.

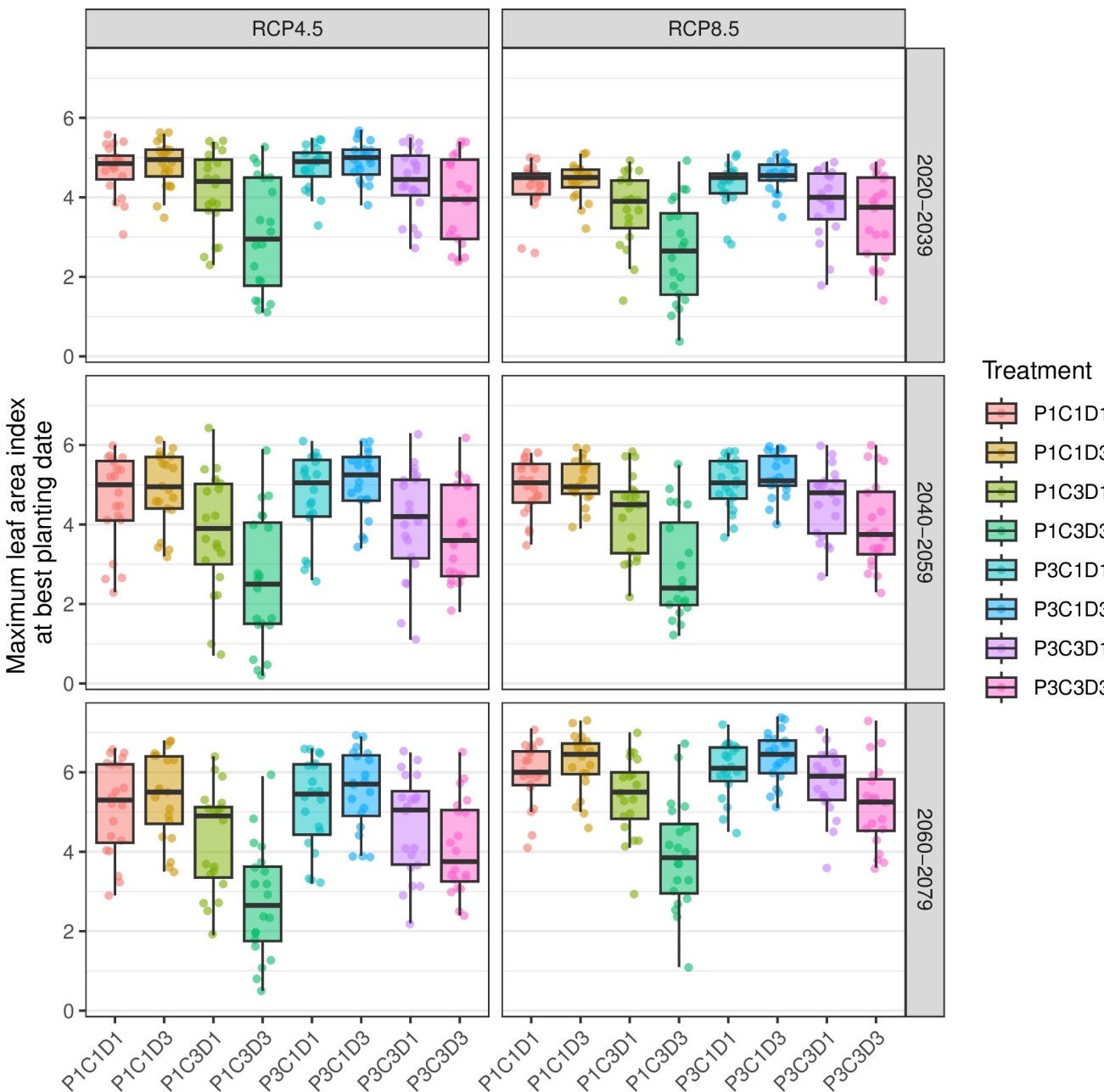

**Fig 4. The impact of the future climate on the maximum leaf area index according to RCP, period, and practices linked to compost fertilization.** Compost applied in March (P1) and in mid-May (P3), Silo compost based on small ruminant manure (sheep and goats) + bush straw (C1), Farmer compost based on small ruminant manure (C3), dose of compost applied of 2.5 t/ha (D1) and 7.5 t/ha (D3). For each boxplot, the points are the individual values (n = 20).

Moreover, the levels of RMSE for the seed cotton yield were similar to those of [49], who calibrated the CSM-CROPGRO-Cotton for three cultivars with RMSE levels from 153 to 448 kg/ha. The data used to simulate the model in our study were considered good according to previous research [61]. According to [62], CSM-CROPGRO-Cotton reveals a simulation of flowering days that is close to observed values of the RMSE, which was less than three days. Similarly [63], found that DSSAT could simulate the capsule ripening period with an RMSE of less than three days.

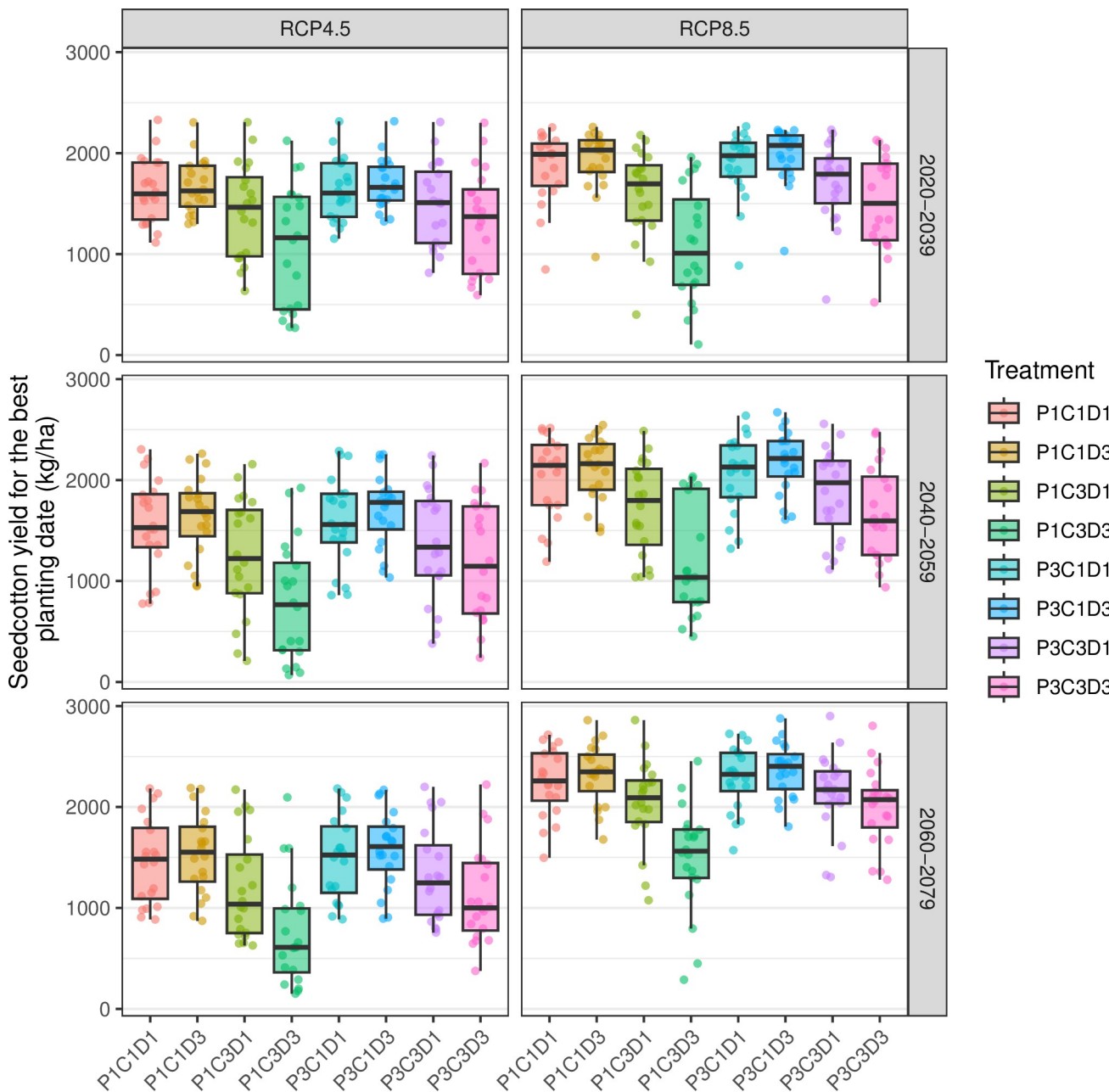

**Fig 5. Impact of future climate on cotton yield according to RCP, period, and practices linked to compost fertilization.** Compost applied in March (P1) and in mid-May (P3), Silo compost based on small ruminant manure (sheep and goats) + bush straw (C1), Farmer compost based on small ruminant manure (C3), dose of compost applied of 2.5 t/ha (D1) and 7.5 t/ha (D3). For each boxplot, the points are the individual values (n = 20).

Our study's pessimistic scenario (RCP8.5) produced better seed cotton yield and performance than the moderate emission scenario (RCP4.5). This corroborates the findings of [64], where an increase in the atmospheric concentration of $CO_2$ had compensated for the expected reductions in seed cotton yield due to high temperature and reduced precipitation [49]. According to [65], cotton photosynthetic activity is proportional to $CO_2$ levels, implying that cotton development is a function of the amount of $CO_2$ in the air. The atmospheric concentration of $CO_2$ up to 650 ppm results in optimum growth for cotton plants [66].

Our study showed increased yields of cotton from small ruminant silo compost compared with farmers' compost. The quality of the fertilizer produced by the farmers needs to be improved to compensate for losses due to mineralization, which falls short of the quality of the products obtained at the research station. According to [26], farmers produce only 6% of the recommended average quality manure or compost today. The quantity of compost is insufficient and may not be increased sufficiently; therefore, its quality has to be improved [26]. The nutrient content of compost depends on the organic raw materials used, compost processing conditions, and composting time [67, 68]. According to [69], for organic manure, not only the litter's quality but also the decomposition duration should be taken into account to have a better yield. A comprehensive simulation of the impact of future climate on organic cotton yields in our study showed changes in the optimistic RCP4.5 scenario with the mid-May application period of small ruminant silo compost at 7.5 t/ha. However, it indicated much lower seed cotton yields with the March and April application periods. This was probably due to decomposition of the compost before incorporation. Organic farming techniques increase soil organic matter and productivity, especially under drought conditions [70]. We found that the latest application rate performed best, similar to [71] in a perennial forage. Near-infrared spectroscopy has successfully estimated organic fertilizer quality [72] and could further enhance the efficiency of compost use in organic farming in Mali.

## 5. Conclusion

The CSM-CROPGRO-Cotton model was reasonably calibrated using observations such as phenology, leaf area index, and yield under organic cotton production in Mali. The application of silo compost at 7.5 t/ha in mid-May, near the beginning of the rainy season in Mali, should be beneficial for organic seed cotton yield, whether under RCP4.5 or RCP8.5 with the future climate scenario of 2020-2079. The crop model could be used to tailor adapted solutions under climate change in Mali for other crops or other management.

## Supporting information

**S1 Dataset.**
(ZIP)

## Acknowledgments

The authors would like to thank the Government of Mali, the cotton program team (Tinamoud CISSE, Mamadou TRAORE, Madelene DEMBELE, Karamoko KAMISSOKO, Yacouba TRAORE), the members of AgrECo and all those who contributed to improving this manuscript.

## Author Contributions

**Conceptualization:** Ginette Dembélé, Romain Loison, Sidiki Gabriel Dembélé, Fagaye Sissoko.

**Data curation:** Ginette Dembélé.

**Funding acquisition:** Fagaye Sissoko.

**Investigation:** Ginette Dembélé, Romain Loison, Amadou Traoré, Sidiki Gabriel Dembélé, Fagaye Sissoko.

**Methodology:** Ginette Dembélé, Romain Loison, Amadou Traoré, Sidiki Gabriel Dembélé, Fagaye Sissoko.

**Project administration:** Fagaye Sissoko.

**Supervision:** Sidiki Gabriel Dembélé, Fagaye Sissoko.

**Visualization:** Romain Loison.

**Writing – original draft:** Ginette Dembélé, Romain Loison.

**Writing – review & editing:** Ginette Dembélé, Romain Loison, Amadou Traoré, Fagaye Sissoko.

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
