## [Decision Letter · Decision Letter 0]

22 Mar 2024

PONE-D-23-33999Optimizing type, date and dose of compost fertilization of organic cotton under climate change in Mali: A modeling studyPLOS ONE

Dear Dr. Loison,

Thank you for submitting your manuscript to PLOS ONE. After careful consideration, we feel that it has merit but does not fully meet PLOS ONE’s publication criteria as it currently stands. Therefore, we invite you to submit a revised version of the manuscript that addresses the points raised during the review process.

We look forward to receiving your revised manuscript.

Kind regards,

Sudeshna Bhattacharjya, Ph.D

Academic Editor

PLOS ONE

Journal Requirements:

AgrECo CML1430 whose financing agreement has been signed between AFD and the Government of Mali

Reviewers' comments:

Reviewer's Responses to Questions

**Comments to the Author**

1. Is the manuscript technically sound, and do the data support the conclusions?

Reviewer #1: Yes

Reviewer #2: Partly

2. Has the statistical analysis been performed appropriately and rigorously? 

Reviewer #1: Yes

Reviewer #2: Yes

3. Have the authors made all data underlying the findings in their manuscript fully available?

Reviewer #1: Yes

Reviewer #2: Yes

4. Is the manuscript presented in an intelligible fashion and written in standard English?

Reviewer #1: Yes

Reviewer #2: Yes

5. Review Comments to the Author

Reviewer #1: The manuscript is very well written and experimental methods as well as the model optimization are well executed. Very minor rectifications are required which are highlighted in the comment section. The study should be published with having those rectification incorporated.

Reviewer #2: Composting is an effective and sustainable technique in organic farming because it helps to reduce nutrient loss through leaching and reduces the need for synthetic fertilizers. It acts as a natural storehouse of both micronutrients like zinc, iron, copper, and manganese, and macronutrients such as nitrogen, phosphorus, potassium, calcium, and magnesium etc. The specific nutrients present depend on the type and composition of the organic material being composted. Additionally, composting provides a reliable method for improving the physical, chemical, and biological qualities of soil in agricultural contexts. Furthermore, applying compost leads to higher crop yields, mainly because it enhances soil fertility and provides essential micronutrients and macronutrients. Numerous studies have highlighted the beneficial effects of compost on the growth and physical characteristics of cotton plants.

Thus, before accepting needs to be modify as per the comments given below:

Some of the study reported that Cotton is C3 and CO2 enrichment enhances yield (Sankaranarayanan et al., 2010) thus specifying the date application of compost enhanced the yield of cotton crop need to be more discussed.

In Introduction, importance of the study needs to be highlighted with more recent references so that study can be justified. Write the importance of this study in global context.

Why DSSATCSM-CROPGRO-Cotton model was used, need more justification as well as include R2 value in both the RCPs. Is this model considered mineralization of compost in soil for prediction?

Whether different sources of composts (elemental composition) were analyzed or not?

In this study, date of sowing of cotton is July 01, 2021, Generally, application of compost 2-3 week before sowing is recommended, but in this study application in mid-May emphasized therefore proper discussion is required indicating to show why application of this model is important rather than going for general recommendation

6. PLOS authors have the option to publish the peer review history of their article (what does this mean?). If published, this will include your full peer review and any attached files.

Reviewer #1: No

Reviewer #2: **Yes: **Rahul Mishra, ICAR-Indian Institute of Soil Science, Bhopal, M.P. India

---

## [Author Response · Author response to Decision Letter 0]

22 Apr 2024

The authors are grateful to the reviewers for their comments, which have helped improve the manuscript. In the revised manuscript with track changes, revisions related to reviewer #1's comments are highlighted in yellow, and those of reviewer #2 are highlighted in blue.

Reviewer #1: 

C1. Line no 64-78: In this paragraph, authors should highlight the urgency or potential lacunas that this study addresses. The title highlights the type, date and dose of organic input, in that direction, authors should highlight that how these factors are currently affecting the seed yield of cotton in Mali with references. The gaps and objectives of this study has to come in this section. 

R1. Thank you. The introduction has been augmented to present better the gap in knowledge and the study's objective.

C2. L32-33. Key words are lengthy. Try to reduce it and give those key words that best describe the article

R2. We modified the keywords, making them shorter and ensuring they describe the content of the article.

C3. L85. Check the journal guidelines. It should be Fig 1 not Figure 1

R3. Thank you. All references to figures have been updated to comply with the journal guidelines.

C4. L87. Check the figure caption style and follow journal guideline

R4. Thank you. All figure captions have been updated to comply with the journal guidelines.

C5. L186. Check the journal guidelines. It should be Fig 2 not Figure 2

R5. Thank you. All references to figures have been updated to comply with the journal guidelines.

C6. L190. Check the figure caption style and follow journal guideline

 R6. Thank you. All figure captions have been updated to comply with the journal guidelines.

C7. L214. Check the abbreviation and use same abbreviation uniformly throughout the manuscript

R7. We replaced NMRSE by nRMSE

C8. L235. Interchange the minimum and maximum words according to the data

R8. We interchanged the min and max to correspond to what they refer to in the text.

C9. L242. Check the solar radiation data, use point instead of comma

R9. We replaced the comma by point. 

C10. L254. Check the journal guidelines. It should be Fig 3 not Figure 3

R10. Thank you. All references to figures have been updated to comply with the journal guidelines.

C11. L258. Use past tense.

R11. We used past tense (had instead of has).

C12. L268. Check the figure caption style and follow journal guideline.

R12. Thank you. All figure captions have been updated to comply with the journal guidelines.

C13. L274. See the citation format.

R13. Thank you, all references to figures have been updated to comply with the journal guidelines.

C14. L282. Modify figure citation according to the journal guideline

R14. Thank you. All references to figures have been updated to comply with the journal guidelines.

C15. L296. Figure caption.

R15. Thank you. All figure captions have been updated to comply with the journal guidelines.

Reviewer #2: 

C16. Some of the study reported that Cotton is C3 and CO2 enrichment enhances yield (Sankaranarayanan et al., 2010) thus specifying the date application of compost enhanced the yield of cotton crop need to be more discussed.

R16. Thank you for your comment. We indeed used the suggested review paper in our introduction to highlight the importance of organic manure like compost in increasing soil organic content in cotton systems under climate change to reduce the impact of climate change. 

C17. In introduction, importance of the study needs to be highlighted with more recent references so that study can be justified. Write the importance of this study in global context.

R17. Thank you very much; please kindly see the response to comment C1 in yellow in the introduction. Some recent publications have been added (2 from 2022, 1 from 2023). We rewrote the end of the introduction to highlight better the importance of this study in the global context.

C18. Why DSSAT CSM-CROPGRO-Cotton model was used, need more justification as well as include R2 value in both the RCPs. Is this model considered mineralization of compost in soil for prediction?

R18. Thank you for your comment. We justified the choice of model and described it better. The details are highlighted in blue in the M&M section 2.3.1. Under both RCPs, this is an in-silico experiment and no observation can be compared to simulated future performance under climate change conditions. Under such conditions, no R2 can be computed. 

C19. Whether different sources of composts (elemental composition) were analyzed or not?

R19. The types of composts were analyzed for elemental composition, a table was added (Table 2).

C20. In this study, date of sowing of cotton is July 01, 2021, Generally, application of compost 2-3 week before sowing is recommended, but in this study application in mid-May emphasized therefore proper discussion is required indicating to show why application of this model is important rather than going for general recommendation

R20. Thank you for your comment. The reference 28 specifies that organic manure is usually applied between March and April in Mali. In addition, an unpublished survey of farmers producing organic cotton in Mali has identified that compost is also commonly applied in March or April but rarely after. Our work highlights the need for a change of practice.

---

## [Decision Letter · Decision Letter 1]

29 May 2024

PONE-D-23-33999R1Optimizing type, date, and dose of compost fertilization of organic cotton under climate change in Mali: A modeling studyPLOS ONE

Dear Dr. Loison,

Thank you for submitting your manuscript to PLOS ONE. After careful consideration, we feel that it still needs some minor revision to fully meet PLOS ONE’s publication criteria as it currently stands. Therefore, we invite you to submit a revised version of the manuscript that addresses the points raised during the review process.

We look forward to receiving your revised manuscript.

Kind regards,

Sudeshna Bhattacharjya, Ph.D

Academic Editor

PLOS ONE

Journal Requirements:

Reviewers' comments:

Reviewer's Responses to Questions

**Comments to the Author**

1. If the authors have adequately addressed your comments raised in a previous round of review and you feel that this manuscript is now acceptable for publication, you may indicate that here to bypass the “Comments to the Author” section, enter your conflict of interest statement in the “Confidential to Editor” section, and submit your "Accept" recommendation.

Reviewer #1: (No Response)

Reviewer #2: All comments have been addressed

2. Is the manuscript technically sound, and do the data support the conclusions?

Reviewer #1: Yes

Reviewer #2: Yes

3. Has the statistical analysis been performed appropriately and rigorously? 

Reviewer #1: Yes

Reviewer #2: Yes

4. Have the authors made all data underlying the findings in their manuscript fully available?

Reviewer #1: Yes

Reviewer #2: Yes

5. Is the manuscript presented in an intelligible fashion and written in standard English?

Reviewer #1: Yes

Reviewer #2: Yes

6. Review Comments to the Author

Reviewer #1: General comments they have worked on which is fin but still the research gap part is lacking. They have added only a paragraph that to not ensuring why application of compost and their timing of application is important for Mali cotton farmers. I understand that application of manure organics undoubtedly improves the soil and crop performance. But it has to come in the manuscript. Without highlighting any potential gap or requirement of this research how we are going to justify that our work is the need of the hour. I request authors to include these things. What is the objective here. What they are trying to address by optimizing the date dose and timing of manure application and how it is going to address the climate change issues. Ultimately, if they address these things then it would be good.

Reviewer #2: Accepted for publication

The study on optimizing the type, timing, and dosage of compost fertilization for organic cotton in Mali under climate change conditions is crucial As climate change alters growing conditions, finding the best composting practices ensures that organic cotton farming remains resilient and sustainable. By determining the optimal composting strategies, farmers can maximize their cotton yields, contributing to economic stability. findings can inform agricultural policies and practices, helping to develop guidelines for compost use under changing climatic conditions,

7. PLOS authors have the option to publish the peer review history of their article (what does this mean?). If published, this will include your full peer review and any attached files.

Reviewer #1: No

Reviewer #2: **Yes: **Rahul Mishra

---

## [Author Response · Author response to Decision Letter 1]

15 Jul 2024

The authors are grateful to the reviewer for his comment. In the revised manuscript with track changes, revisions are highlighted in green.

Reviewer #1: 

C1. General comments they have worked on which is fin but still the research gap part is lacking. They have added only a paragraph that to not ensuring why application of compost and their timing of application is important for Mali cotton farmers. I understand that application of manure organics undoubtedly improves the soil and crop performance. But it has to come in the manuscript. Without highlighting any potential gap or requirement of this research how we are going to justify that our work is the need of the hour. I request authors to include these things. What is the objective here. What they are trying to address by optimizing the date dose and timing of manure application and how it is going to address the climate change issues. Ultimately, if they address these things then it would be good.

R1. Thank you. The introduction has been augmented a second time to present better the gap in knowledge and the study's objective. Please kindly check the introduction.

---

## [Decision Letter · Decision Letter 2]

30 Jul 2024

Optimizing type, date, and dose of compost fertilization of organic cotton under climate change in Mali: A modeling study

PONE-D-23-33999R2

Dear Dr. Romain Loison,

We’re pleased to inform you that your manuscript has been judged scientifically suitable for publication and will be formally accepted for publication once it meets all outstanding technical requirements.

Kind regards,

Sudeshna Bhattacharjya, Ph.D

Academic Editor

PLOS ONE

Additional Editor Comments (optional):

Reviewers' comments:

Reviewer's Responses to Questions

**Comments to the Author**

1. If the authors have adequately addressed your comments raised in a previous round of review and you feel that this manuscript is now acceptable for publication, you may indicate that here to bypass the “Comments to the Author” section, enter your conflict of interest statement in the “Confidential to Editor” section, and submit your "Accept" recommendation.

Reviewer #1: All comments have been addressed

2. Is the manuscript technically sound, and do the data support the conclusions?

Reviewer #1: Yes

3. Has the statistical analysis been performed appropriately and rigorously? 

Reviewer #1: Yes

4. Have the authors made all data underlying the findings in their manuscript fully available?

Reviewer #1: Yes

5. Is the manuscript presented in an intelligible fashion and written in standard English?

Reviewer #1: Yes

6. Review Comments to the Author

Reviewer #1: The authors have updated the manuscript quite satisfactorily.

The manuscript is publication ready.

Thank you.

7. PLOS authors have the option to publish the peer review history of their article (what does this mean?). If published, this will include your full peer review and any attached files.

Reviewer #1: No

---

## [Editor Report · Acceptance letter]

5 Aug 2024

PONE-D-23-33999R2 

PLOS ONE

Dear Dr. Loison, 

I'm pleased to inform you that your manuscript has been deemed suitable for publication in PLOS ONE. Congratulations! Your manuscript is now being handed over to our production team.

Kind regards, 

on behalf of

Dr. Sudeshna Bhattacharjya 

Academic Editor

PLOS ONE